# Clinical and molecular correlation defines activity of physiological pathways in life-sustaining kidney xenotransplantation

Daniel J. Firl [1,2,3], Grace Lassiter[1,3], Takayuki Hirose [1], Robert Policastro[2], Ashley D'Attilio[1], James F. Markmann [1], Tatsuo Kawai [1,4] & Katherine C. Hall [2,4] ✉

Porcine kidney xenotransplantation is accelerating towards clinical translation. However, despite the demonstrated ability of porcine kidneys to remove metabolic waste products, questions remain about their ability to faithfully recapitulate renal endocrine functions after transplantation. Here we analyze xenograft growth and function of two kidney dependent endocrine pathways in seventeen cynomolgus macaques after kidney xenotransplantation from gene edited Yucatan minipigs. Xenograft growth, the renin-angiotensinogen aldosterone-system, and the calcium-vitamin D-parathyroid hormone axis are assessed using clinical chemistries data, renin activity and beta-C-terminal-telopeptide assays, kidney graft RNA-sequencing and serial ultrasonography. We demonstrate that xenografts transplanted from minipigs show only modest growth and do not substantially contribute to recipient RAAS pathway activity. However, parathyroid hormone-independent hypercalcemia and hypophosphatemia are observed, suggesting a need for close monitoring and timely intervention during human testing. Further study of these phenotypes is warranted in designing prospective clinical trials.

Chronic kidney disease (CKD) affects 8–16% of the population worldwide and often goes underappreciated until it progresses to end stage kidney disease (ESKD)[1]. ESKD patients with refractory symptoms require lifelong kidney replacement, either dialysis or transplantation. Dialysis extends life but dialysis patients have worse outcomes, poorer quality of life, and increased costs as compared to transplantation[2,3].

Transplantation provides a potentially curative therapy but demand for donor organs far exceeds current levels of donation. One approach to alleviating this organ shortage is xenotransplantation of porcine kidneys[4]. Porcine kidneys are similar in size, shape, and function to human kidneys, are ethically acceptable and have been used for decades in preclinical non-human-primate (NHP) models of kidney xenotransplantation. The application of clustered regularly interspaced short palindromic repeats (CRISPR)/CRISPR-associated protein

9 (Cas9) has accelerated efforts to modify the porcine genome and facilitate xenotransplantation[5]. In NHP recipients, gene-edited porcine donors have enabled survival of >500 days[6] and multiple groups have now demonstrated lack of hyperacute rejection in recently deceased humans[7,8]. Given this promising survival data, a new level of scrutiny is warranted and questions about which kidney functions a porcine donor organ can or cannot replace have become more relevant.

Filtration of nitrogenous waste products is a principal and necessary function of the kidney to sustain life. However, kidneys are complex and also participate extensively in various endocrine pathways[1]. This study was designed to evaluate these secondary functions of kidney xenografts in a cohort of long-term surviving NHPs to anticipate complications and inform future clinical trials of porcine xenografts. Specific topics of interest included: (1) porcine xenograft growth[9,10], (2) porcine xenograft participation in

[1]Center for Transplantation Sciences, Massachusetts General Hospital, Boston, MA, USA. [2]eGenesis Inc, Cambridge, MA, USA. [3]These authors contributed equally: Daniel J. Firl, Grace Lassiter. [4]These authors jointly supervised this work: Tatsuo Kawai, Katherine C. Hall. ✉e-mail: Katherine.hall@egenesisbio.com

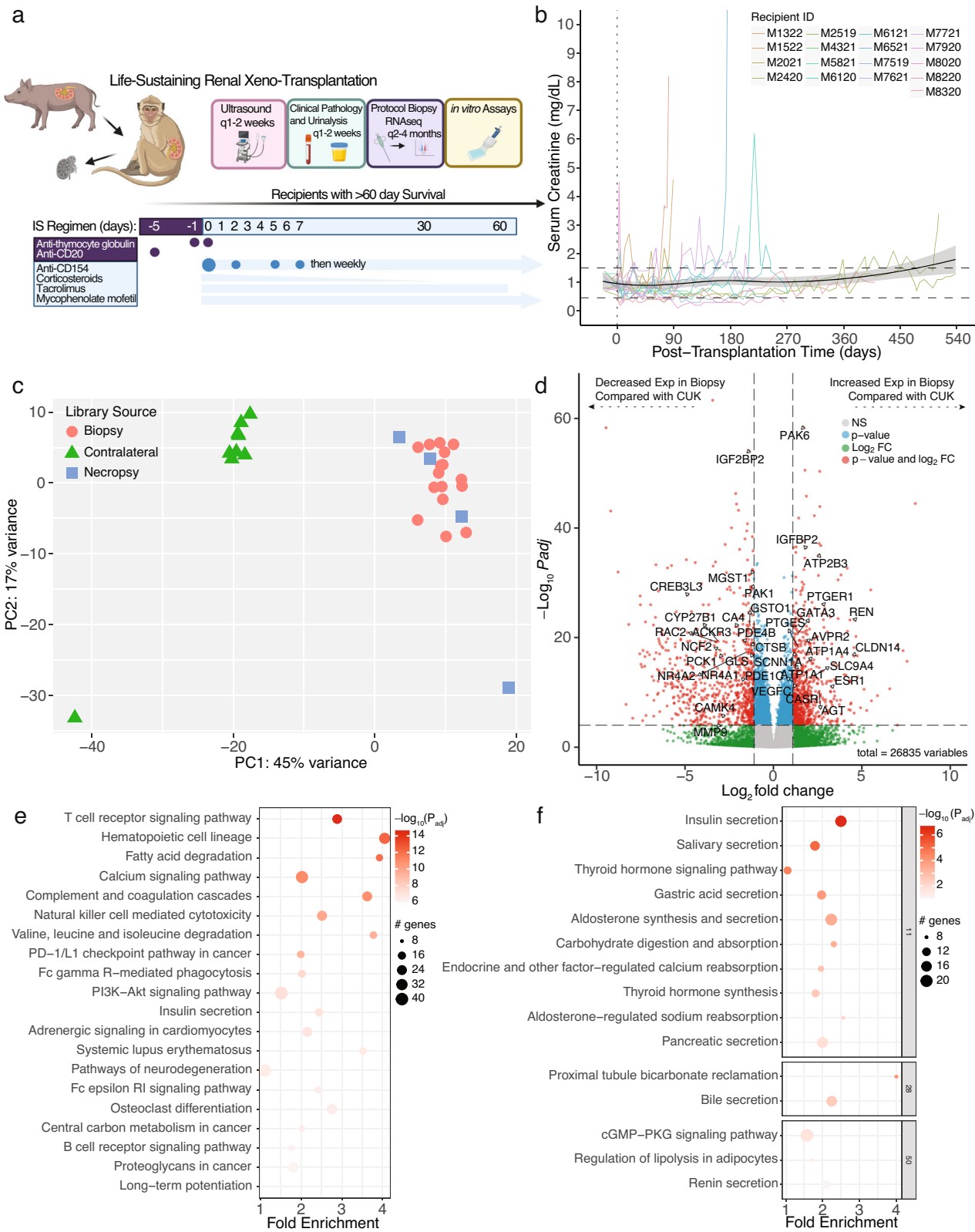

NHP renin-angiotensin-aldosterone-system (RAAS) signaling[11–13], and (3) porcine xenograft maintenance of physiological levels of blood electrolytes by both tubular reabsorption and participation in the calcium-vitamin-D-parathyroid-hormone axis[10,14]. Here we show minimal xenograft growth, lack of xenograft involvement in the RAAS pathway, and PTH-independent hypercalcemia and hypophosphatemia after kidney xenotransplantation.

## Results

### Long-term survivors after kidney xenotransplantation provide clinical and molecular insight

Gene-edited Yucatan minipig were used as kidney donors for xenotransplantation to cynomolgus macaques and 17 recipients with survival >60 days were analyzed for this study (Fig. 1a; Supplementary Tables 1, 2). Measures of renal filtration were generally normal except

**Fig. 1 | Study overview and introduction to molecular profiling of tissue samples. a** Schematic overview of experimental life sustaining pig to NHP kidney xenotransplantation (created with Biorender.com). **b** Serum creatinine measured from each animal over time ($n = 17$ biologically independent transplants). Black solid line is the LOESS estimate, a non-parametric regression method estimating the relationship between creatinine and PTT across all studies; gray shaded region is 95% confidence interval for the estimate. Dashed lines represent normal range as provided by the Center for Comparative Medicine from MGH. **c** Principal components analysis of porcine RNA-seq libraries (aligned transcripts to porcine genome; NHP aligned transcripts discarded) from each contralateral kidney (green triangles, $n = 9$ biologically independent samples), biopsy kidney tissue (red circles, $n = 16$ biologically independent samples), and necropsy kidney tissue (blue squares, $n = 4$ biologically independent samples). **d** Volcano plot of differentially expressed porcine genes comparing biopsy tissue samples to contralateral untransplanted kidney tissues (necropsy samples not included). Color coding: gray = no statistically significant difference and not differentially expressed; green = differentially expressed but not statistically significantly different; blue = statistically significantly different though not significantly differentially expressed; red = statistically

significantly differentially expressed and statistically significantly different from contralateral untransplanted expression. **e** Top 10 enriched pathways by "pathfindR" pathway analysis using KEGG pathways, ordered by −log10(Padj). **f** Subnetwork analysis and clustering performed to group enriched pathways together if driven by similar genes. Statistical testing for panel d is performed by the DESeq2 tool (refer to materials and methods) using the Wald test to calculate p-values for each gene, which are then adjusted for multiple comparisons using the Benjamini-Hochberg method to control the false discovery rate (FDR-adjusted, two-sided, $p$-values < 0.05 were considered differentially expressed). The log2 fold change (LFC) is also calculated for each gene, which represents the difference in expression between the two conditions on a log2 scale. Statistical testing for panels e and f are performed by pathfindR using the hypergeometric test on the FDR-adjusted results from DESeq2 for multiple comparisons. CUK contralateral untransplanted kidney, Exp expression, LOESS locally estimated scatterplot smoothing, MGH Massachusetts General Hospital, Padj FDR-adjusted $p$-values using the Benjamini-Hochberg method, calculated by DESeq2, PC principal component.

in the early postoperative period or just preceding graft loss (Fig. 1b). Tissue biopsies from stable transplants without histological evidence of rejection were subjected to RNA-seq and bioinformatically processed to retain only porcine transcripts. These libraries generally clustered by sample source on unsupervised principal components analysis (contralateral untransplanted kidney (CUK), biopsy, necropsy; Fig. 1c). Comparing biopsy vs CUK samples, 1742 porcine genes were differentially expressed ($P_{adjusted} < 0.05$; 847 Log2-fold-change [LFC] > 0.5 and 895 with LFC < −0.5; Fig. 1d). Pathway analysis identified several "immune-cell-related" gene sets amongst DEG (differentially expressed genes) networks (7/20 top pathways, Fig. 1e). After clustering the subnetwork components, it was possible to identify enriched pathways related to kidney endocrine functions; though enrichment scores were more modest compared to the top pathways (Fig. 1e, f).

## Minimal xenograft growth after kidney xenotransplantation

Kidney size was measured serially as long-axis length on sagittal plane ultrasonography (Fig. 2a). Early after xenotransplantation, growth may be confounded by hydronephrosis which was observed in 11/17 animals, typically mild (SFU-II) and self-limited (resolved ~2 weeks post-transplantation time (PTT)); consistent with a non-obstructive process (Supplementary Table 3). Despite hydronephrosis, for both absolute length and relative change, there was no obvious trend over 17 experiments (Fig. 2b, c). Fixed effects modeling revealed a modest though statistically significant increase of xenograft size over time as both absolute (0.04 cm per month, $P < 0.001$; Supplementary Table 4) and relative measures (0.7% per month, $P < 0.001$; Supplementary Table 5).

## Porcine xenograft does not efficiently initiate cynomolgus macaque RAAS

To evaluate contribution of porcine xenografts to RAAS in vivo, plasma renin activity (PRA), measured as in vitro generated Angiotensin I (AngI), was assayed pre- and post-xenotransplantation (Fig. 3b). While the absolute magnitude of AngI generated in the PRA assay varied significantly preoperatively (500–1400 pg/mL/6 h), consistent and statistically significant decreases following xenotransplantation were observed (AngI was ~7.1% of the pre-transplant level by 30–40 days PTT and ~4.3% by 80–90 days PTT; Fig. 3b and Supplementary Fig. 1a, b). M4321 and M6521, recipients that retained a single native kidney at transplant, showed increased PRA early after transplant, consistent with renin release from the remaining NHP kidney. In both cases, after completion of native nephrectomy (~20 days PTT), a decrease in PRA was observed, consistent with other recipients (Supplementary Fig. 1a, b). Aldosterone levels were significantly lower after transplant,

consistent with the PRA results (Fig. 3c and Supplementary Fig. 1c). Levels of urinary sodium did not suggest a significant concentrating defect after xenotransplantation (Supplementary Fig. 6e). RNA-seq identified "RAAS-related" pathways as significantly enriched in biopsies compared with CUK (Fig. 3d, e). Downstream effectors of RAAS signaling, such as transcription factor *NR4A2* were decreased in expression, whereas cell surface ion channels repressed by RAAS such as *SCNN1, KCNJ1, KCNK3/9*, were increased. Regulators of renin, such as *PRKG2*, were downregulated. *AGT* and *REN* were amongst the most upregulated transcripts (LFC = 2.5 and 4.5, respectively; $P_{adjusted} < 5E−7$ and $<9E−18$, respectively; Fig. 1e), suggesting reduced negative feedback inhibition. Taken together, data from these complementary techniques strongly suggest absence of efficient initiation of NHP RAAS by porcine-derived renin. To exclude the possibility that the decrease in renin activity was due to the surgical procedure or immunosuppression agents, renin activity was assayed in four recipients of life-sustaining allotransplantation on an anti-CD154 based immunosuppression (Supplementary Fig. 2a, b)[15]. Substantial variation in the absolute level of renin activity is noted among the individuals, but as either an absolute or relative measure, significant renin activity is observed post transplantation in strong contrast to the xenotransplanted NHPs (Supplementary Fig. 3a, b).

## Hypercalcemia and hypophosphatemia are common after xenotransplantation

In xenotransplantation, the porcine kidney must regulate concentrations of electrolytes, including sodium, potassium, calcium, and phosphate. While sodium and potassium were generally within normal ranges (Supplementary Fig. 4a, b), calcium and phosphorus were not (Fig. 4a, b). By 30 days PTT, 82% (14/17) of NHPs had ≥1 measured calcium above the normal range. In most recipients, hypercalcemia was mild-to-moderate (~11.5–12.0 mg/dL) but 35% (6/17) had ≥1 instance of severe hypercalcemia (>14 mg/dL). As a point of comparison, chemistry data from four allotransplanted cynomolgus macaques with an anti-CD154-based immunosuppression was examined (Supplementary Fig. 2a, b)[15]. These animals were maintained with life-sustaining function for >180 days PTT and demonstrated relatively normal serum levels of calcium and phosphorus (Supplementary Fig. 5a–c). These data support the notion that in this regard xenograft function does differ from allograft. In the allografts only, anti-CD154 was weaned starting after days 200 PTT and the borderline hypophosphatemia observed in the last time points of two of these transplants may reflect tubular dysfunction from inflammatory processes due to IS reduction.

Urinary calcium was measured in a subset of animals (Supplementary Table 1, Fig. 4c and Supplementary Fig. 6a, b). Individual animals show a trend of modestly decreased urine calcium immediately

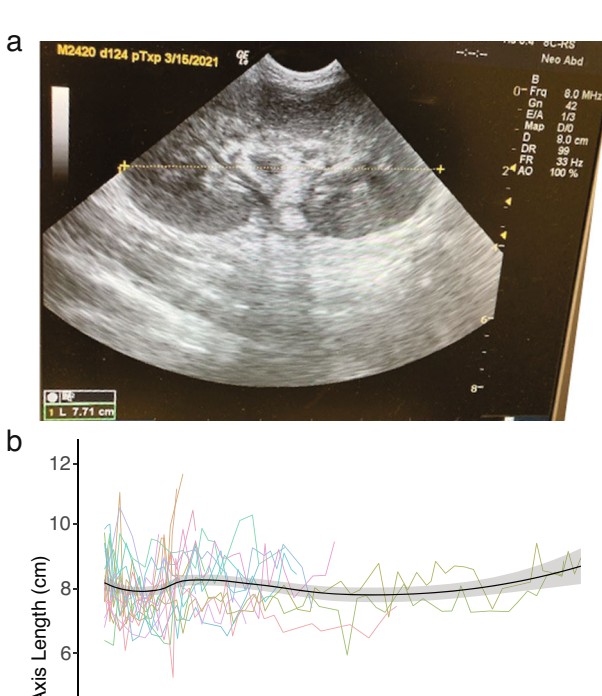

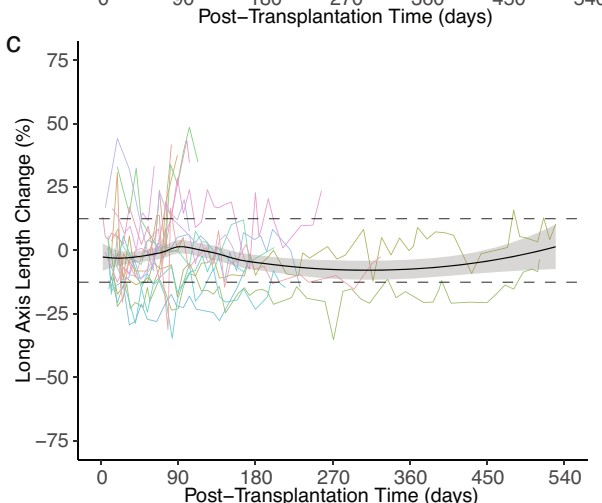

**Fig. 2 | Longitudinal analysis of xenograft length after life-sustaining kidney xenotransplantation shows minimal growth. a** Example ultrasound (US) of NHP M2420 xenograft at 124 days PTT. **b** Individual animals' absolute changes in kidney graft length over time. Black solid line is the LOESS estimate, a non-parametric regression method estimating the relationship between absolute length and PTT across all transplants; gray shaded region is 95% confidence interval for the estimate. **c** As in (**b**) but for relative change in length over time. Dashed lines represent estimated between US variation based on ultrasounds performed on the same animal in the same week. *n* = 17 biologically independent transplants for panels **b**, **c**. LOESS locally estimated scatterplot smoothing.

post-transplantation (Supplementary Fig. 6f) and statistical analysis on binned normalized urinary calcium, accounting for repeated measures, demonstrated a non-significant trend towards decreased normalized urine calcium to creatinine ratio during the period between 0 and 30 days PTT (*P* = 0.70, Fig. 4c, and Supplementary Fig. 6a, b). However, in

the context of hypercalcemia, values of urine calcium would be expected to be significantly increased if the porcine kidney was solely sieving blood, thus normal or low urine calcium is consistent with renal retention of calcium. While xenokidneys appeared to retain calcium, there was no objective data to support complications related to this biochemical phenomenon (e.g., nephrocalcinosis), however, limitations of the NHP model prevent assessment of more common and subjective features of hypercalcemia such as mood disorders, fatigue, or muscle weakness.

Concurrently with changes in calcium, levels of serum phosphorus decreased early after transplantation (Fig. 4b). Eighty-eight percent of animals (15/17) had phosphorus below the lower limit of normal by day 30 PTT. Most hypophosphatemia was moderate, with the LOESS (locally estimated scatterplot smoothing) line around 2 mg/dL; but 47% (8/17) experienced ≥1 instance of severe hypophosphatemia (<1 mg/dL). Urinary phosphorus was measured in a subset of animals (Fig. 4d and Supplementary Fig. 6c, d). Individual animal trends indicated there was a modest increase in urinary phosphorus immediately post-transplantation (Supplementary Fig. 6g). Statistical analysis on binned data demonstrated a statistically significant increase in urine phosphorus to urine creatinine ratio in the PTT 0-30 day period (Fig. 4d; *P* < 0.001). Despite the robust observation of hypophosphatemia, complications or adverse events associated with hypophosphatemia, such as red blood cell lysis or respiratory insufficiency were not observed.

RNA-seq revealed DEG related to endocrine regulation of calcium reabsorption, calcium signaling, and proximal tubule bicarbonate reclamation (Fig. 4e). In addition to gene networks, expression of genes of interest were compared, especially those related to paracellular calcium reabsorption (Supplementary Fig. 7). A dramatic increase in *CLDN14* expression (LFC = 4.4; $P_{adjusted}$ < 4E−13) and more modest increases in *CaSR* expression (LFC = 1.3; $P_{adjusted}$ < 1E−4) were noted. With respect to transcellular calcium reabsorption, no change in *TRPV5* expression was observed (LFC = 0.2; $P_{adjusted}$ = 0.64) but there was a significant increase in *CALB1* (LFC = 2.3; $P_{adjusted}$ < 1E−4) which encodes a cytosolically expressed protein critical to mobilizing $Ca^{2+}$ to the basolateral surface of renal epithelial cells. Overall, increases in genes encoding proteins important for handling increased calcium load were observed, though the exact driver of this phenotype remains unresolved.

### Recipient calcium-vitamin D-PTH axis and bone are not sources of dysregulated calcium and phosphorus

ESKD patients typically have hypocalcemia and hyperphosphatemia with resulting hyperparathyroidism. In contrast, NHPs in the preclinical model were healthy before study start. Despite this limitation, the status of components of this axis (Fig. 5a) after xenotransplantation may provide insight into potential clinical ramifications. Calcifediol, calcitriol, and PTH were measured before and after xenotransplantation (Fig. 5b, c, d). Calcifediol and calcitriol remain in approximately the same range with modest variability (Fig. 5b, c and Supplementary Fig. 8a, b). In contrast, PTH was suppressed in 100% (6/6) of animals after transplant (Fig. 5d and Supplementary Fig. 8c; *P* < 0.001) as an appropriate response to hypercalcemia and no PTHrP was ever detected (Supplementary Fig. 8d). Parathyroid and thyroid tissue from a single long-term survivor (M2519, 511 days) were evaluated histopathologically and no evidence of any diagnostic abnormality, such as hypo/hyperplasia, was reported.

While serum levels of calcifediol and calcitriol were not significantly altered, transcriptional level changes in this pathway were noted (Supplementary Fig. 7). *CYP27B1*, responsible for converting calcifediol to calcitriol, was significantly decreased in biopsy vs CUK (LFC = −3.1; $P_{adjusted}$ < 2E−13). Given the relative stability of calcitriol and persistent hypercalcemia, decreased expression of *CYP27B1* likely reflects low PTH levels.

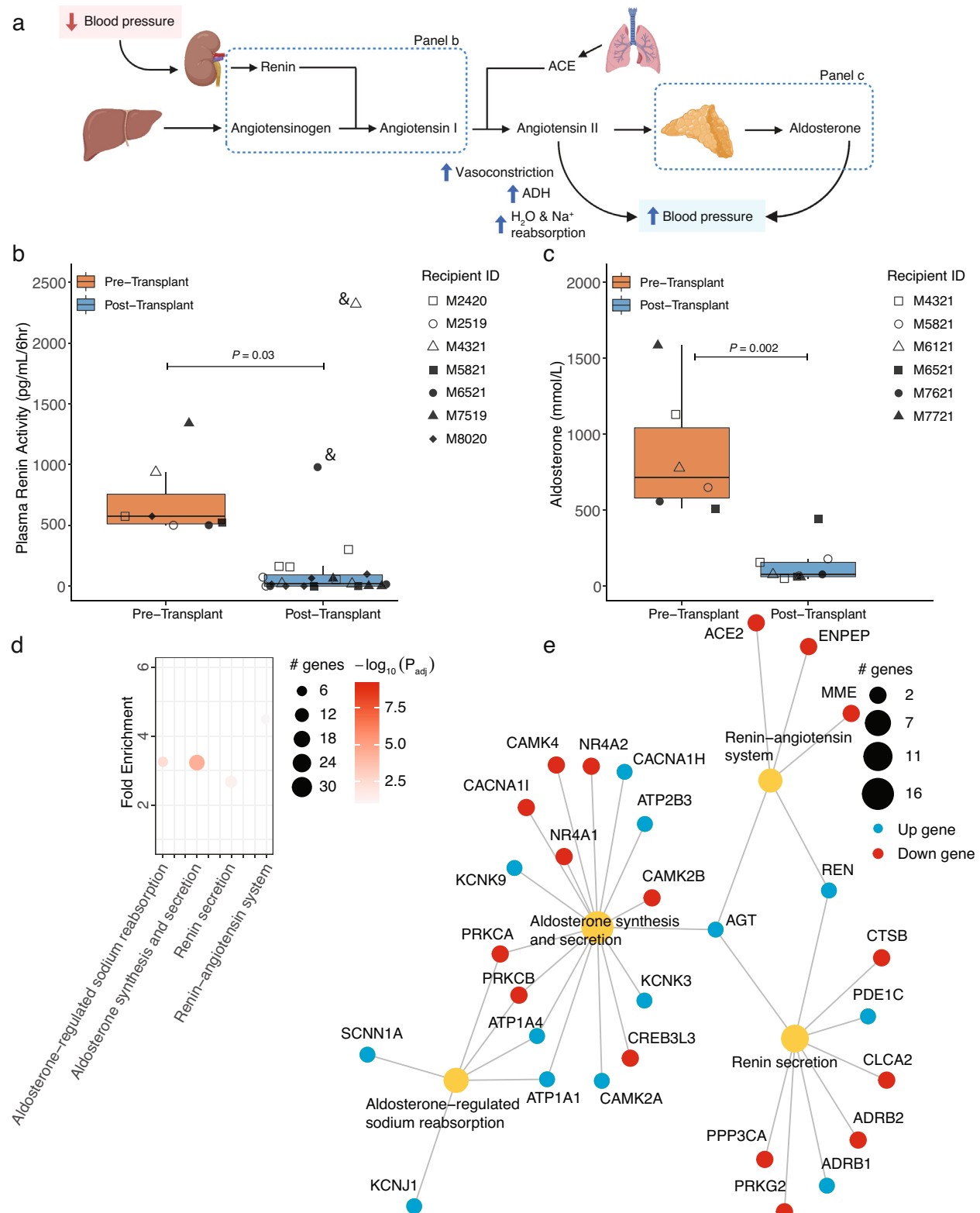

The possibility of aberrant bone turnover as a source of hypercalcemia was investigated. Samples from seven recipients and two pools of cynomolgus samples were assayed for beta-C-terminal telopeptide (CTx), a reference marker of bone resorption. While pre-transplant CTx levels were greater than ranges reported in humans, a robust decrease after transplantation which mirrored the observed trends in PTH was observed (Fig. 5e and Supplementary

Fig. 8e). This suggests that bone resorption may not be responsible for hypercalcemia.

## scRNA-seq confirms expected localization of transcriptional signals from kidney

In addition to bulk RNAseq, tissue samples were subjected to single-cell RNA sequencing (scRNA-seq). 30,161 cells passed quality control

**Fig. 3 | Reduced renin-angiotensin-aldosterone system activity after kidney xenotransplantation. a** Schematic of renin-angiotensin-aldosterone pathway (created with Biorender.com). **b** Plasma renin activity (measured by Angiotensin I generated) was measured in pre- and post-transplant samples from NHP recipients. **c** As in (**b**) but for aldosterone. **d** Pathway analysis of tissue biopsy RNA-seq as compared to CUK demonstrates significantly enriched differential expression in RAAS-related pathways. **e** Gene network mapping differentially expressed genes across gene sets of interest. &Denotes animal with a single native NHP kidney in situ at time this sample was taken, subsequent samples demonstrated consistently low values. For box and whisker plots, the central line in each box represents the median of the distribution, i.e., the value that separates the lower 50% of observations from the upper 50%. The box itself represents the interquartile range (IQR),

which spans from the 25th to the 75th percentile of the distribution. The lower bound of the box is the first quartile (Q1), while the upper bound is the third quartile (Q3). The whiskers of the box and whisker plot extend from the box to the minimum and maximum observations within 1.5 times the IQR of the lower and upper quartile, respectively (panels **b** and **c**). Statistical testing for panels **b** and **c** was performed using generalized linear models with xenotransplantation bin (pre vs post) and recipient IDs as factor variables. Statistical testing for panels **d** and **e** was performed by pathfindR using the hypergeometric test on results from DESeq2 which were FDR-adjusted for multiple comparisons. $n = 7$ biologically independent transplants for panel **b** and $n = 6$ biologically independent transplants for panel **c**. ACE angiotensin converting enzyme, ADH antidiuretic hormone, IQR inter-quartile range (25th–75th percentiles).

from which at least 10 distinct parenchymal cell types were identified based on published marker gene expression patterns (Fig. 6a). To appreciate variation and consistency of this scRNA-seq dataset, *CDH1* and *CDH2* were mapped across cells. In general, past reports have found greater expression of *CDH1* in distal nephron including descending limb/collecting ducts but more *CDH2* in proximal nephron including podocytes and proximal tubule cells, which was confirmed for both proportion of cells expressing (Fig. 6b) and normalized rlog counts (Fig. 6c). *PECAM1* and *RGS5* generally marked endothelial cell types and pericytes and podocytes, respectively. In addition, various transcripts of interest to the physiological pathways above were interrogated. *REN* was localized specifically to pericytes. *CaSR* was highest expressed in thick ascending limb (TAL) but had some additional expression detected in collecting ducts. *CLDN14* was highest expressed in TAL and distal convoluted tubule. *CALB1* was highest expressed in distal convoluted tubule and collecting duct.

## Discussion

Pig to human kidney xenotransplantation is one of the most promising solutions to the donor organ shortage. Gene edits with the potential to improve human compatibility of porcine organs have been made possible by CRISPR/Cas9 technology. The combination of successful gene editing and next-generation immunosuppression has enabled long-term survival after xenotransplantation in NHPs which motivates additional questions about safety and performance of endocrine functions. Analysis of the largest group of long-term survivors reported to date demonstrated only very modest growth of the xenograft, inefficient participation of the porcine organ in the recipient RAAS pathway, and a remarkably reproducible PTH-independent hypercalcemia and hypophosphatemia. These findings enable continued development and thoughtful clinical trial design for first in human (FIH) clinical trials with gene-edited porcine organs.

This study adds molecular data to clinical phenotypes after life-sustaining kidney xenotransplantation. Bulk and scRNA-seq on tissue samples from CUKs and post-transplant biopsies were processed and transcripts which aligned to Sus Scrofa 11.1 were analyzed while those of NHP were discarded, enabling analysis of donor-specific transcriptomic changes. Pathway analysis based on these RNA-seq data revealed that many of the top pathways altered after xenotransplantation were related to immune cells (Fig. 1e). Some of these findings might be partially explained by loss of porcine immune cells, while others may be porcine-specific responses to xenotransplantation. While many of the top altered pathways seemed "immune-related", subnetwork clustering demonstrated enriched kidney endocrine pathways of interest: "Endocrine and other factor-regulated calcium reabsorption", "Aldosterone-regulated sodium reabsorption", and "Renin secretion" (Fig. 1f). Further, scRNA-seq of ~30,000 cells revealed that DEGs were localized to the expected compartments within the kidney (i.e., *REN* in pericytes, *CaSR* in TAL) rather than aberrant expression in other cell types (Fig. 6).

In this study group, kidney length measurement by serial ultrasonography revealed no gross trends when analyzed as absolute kidney

length, or percent kidney length change (Fig. 2b, c). Closer examination with fixed effects modeling revealed a small, though statistically significant, increase in kidney length (Supplementary Tables 4 and 5). Extrapolating this model's estimate suggests an average ~8.0% increase in kidney length per year post transplant. This finding contrasts with previous reports in the literature. In 2000, hDAF transgenic porcine kidneys transplanted into cynomolgus macaques grew in length as much as 30–50% by end of study[9]. While this study had early and universal rejection (maximum survival 78 days), growth has also been observed in more recent studies. Tanabe et al. transplanted *GGTA1* knockout kidneys into baboons and noted expansion of xenokidney volume, ~2–2.5x (maximal survival 193 days)[16]. Finally, a group at the University of Alabama published kidney growth data for a series of six baboons, using varying genotype porcine donors from United Therapeutics[17,18]. Kidney length increased up to ~100% over 20 weeks post-transplant. Interestingly, those authors noted that two of their animals had hydronephrosis which was correlated with growth and that graft size significantly decreased after performing redo ureterovesical anastomosis. In this report, while mild hydronephrosis was common early after transplantation, growth was relatively minor. It is notable that transplants in this study that ended earlier (60–120 days) more often displayed a modest growth phenotype (20–50% length increase; Fig. 2c). This suggests one possible explanation for the difference between this study and previously published results is the selection of animals with survival >60 days. This selection event might enrich for animals with fewer technical issues, less parenchymal edema, less severe hydronephrosis, less early rejection, and/or graft dysfunction, thus less "growth." Another potential explanation might be the strain of pig used. This report uses Yucatan minipigs (mature body weight ~75 kg), while hDAF pigs were generated from Large-White/Landrace hybrid animals (mature body weight ≥300 kg) and United Therapeutics/Revivicor utilizes a German-Landrace breed with a growth hormone receptor knockout gene edit for their 10GE pigs™[19,20]. In summary, in stable NHPs after life-sustaining xenotransplantation, only modest increase in xenokidney length was observed which should not impede clinical translation. Whether this finding would persist beyond 500 days remains unclear with this dataset, but we suggest careful monitoring with scheduled serial ultrasonography for clinical studies.

Amongst this cohort of long-term survivors, functionally relevant RAAS components were measured using orthogonal techniques which demonstrated that porcine xenokidneys do not participate in NHP RAAS. In response to decreased blood pressure, decreased sodium, and/or sympathetic nervous system activation, kidney pericytes secrete renin which cleaves angiotensinogen (AGT) initiating RAAS. After xenotransplantation, there was a significant decrease in plasma renin activity (measured by AngI formation) as well as aldosterone, which could be delayed but not entirely prevented when a single native kidney was left in place and later removed (Fig. 3b, c). RNA-seq revealed DEGs consistent with a compensatory response to lack of RAAS signaling (Fig. 3d, e). Interestingly, *AGT* was amongst the most upregulated transcripts (LFC = 2.5; $P_{adjusted} < 10E-5$). While the liver is the usual source of homeostatic angiotensinogen, this finding supports the

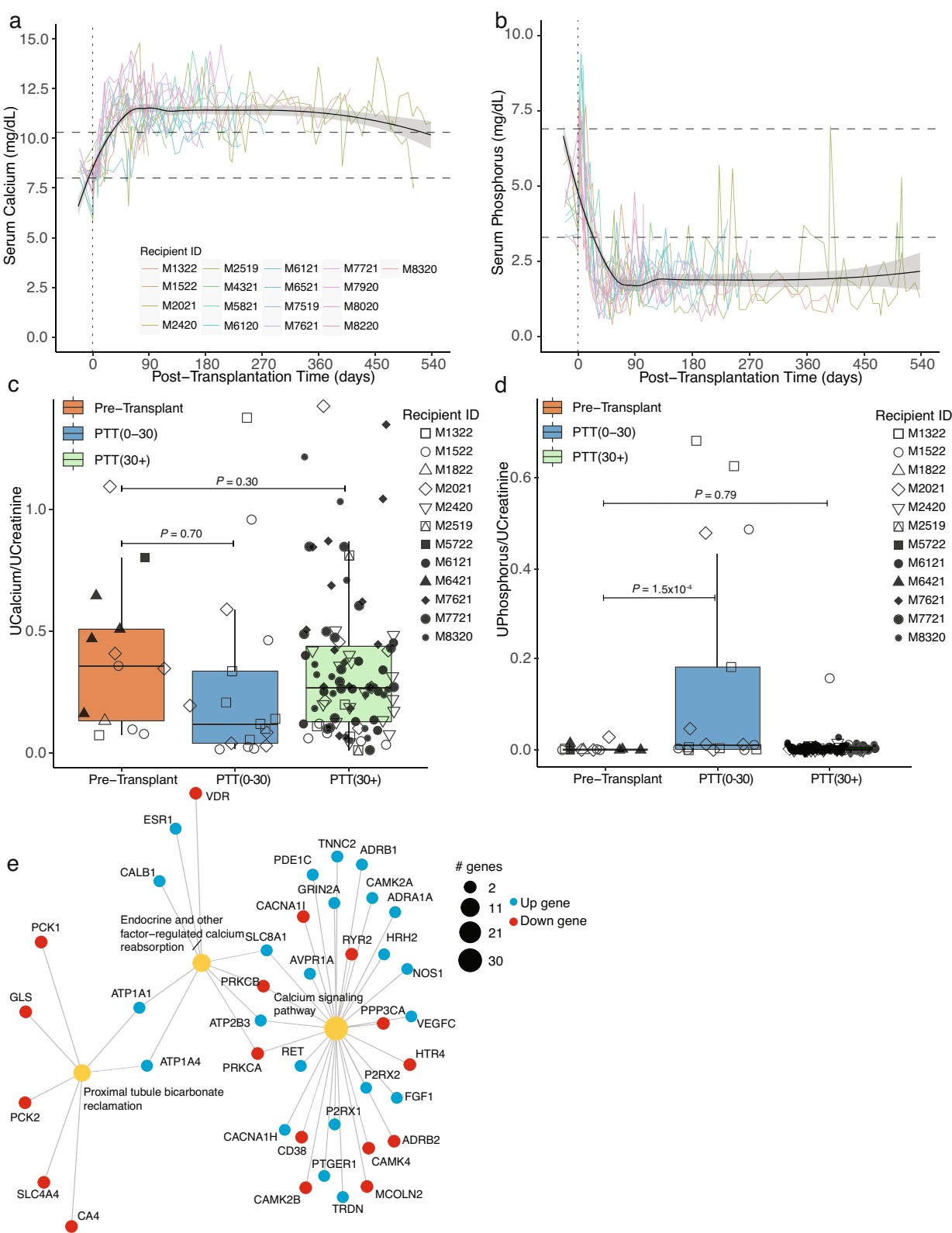

possibility of local, intrarenal RAAS signaling. These data are generally consistent with prior observations in the field that human angiotensinogen is not efficiently cleaved by porcine renin in vitro[11,12,21]. In vivo, baboons experienced episodes of hypovolemia and hypotension with modestly elevated serum creatinine after life-sustaining kidney xenotransplantation, which was corrected via intravenous administration of bolus fluids with no signs of rejection[22]. Navar et al. presented findings

that baboons maintain normal levels of plasma renin (plasma renin activity was not reported) and aldosterone, while angiotensinogen is increased post xenotransplantation. Furthermore, Angiotensin-II was detectable, though at decreased levels, suggesting porcine renin may cleave some baboon AGT[13]. Another interpretation is that intrarenal RAAS can lead to production of Angiotensin-II which acts on the adrenal glands to produce aldosterone. While in this study, RNA-seq

**Fig. 4 | Altered levels of calcium and phosphorus after xenotransplantation.**
**a** Serum calcium was measured longitudinally in 17 recipients. Black solid line is the LOESS estimate, a non-parametric regression method estimating the relationship between calcium and PTT across all transplants; gray shaded region is 95% confidence interval for the estimate. **b** As in (**a**) but for serum phosphorus. Dashed lines represent normal range as provided by the Center for Comparative Medicine, MGH. **c** Urine calcium was measured pre- and post-transplant and normalized to urinary creatinine. **d** As in (**c**) but for urine phosphorus. **e** Gene network analysis mapping differentially expressed genes across gene sets of interest. For box and whisker plots, the central line in each box represents the median of the distribution, i.e., the value that separates the lower 50% of observations from the upper 50%. The box itself represents the interquartile range (IQR), which spans from the 25th to the 75th percentile of the distribution. The lower bound of the box is the first quartile (Q1),

while the upper bound is the third quartile (Q3). The whiskers of the box and whisker plot extend from the box to the minimum and maximum observations within 1.5 times the IQR of the lower and upper quartile, respectively (panels **c** and **d**). Statistical testing for panels **c** and **d** was performed using generalized linear models with xenotransplantation bin (pre vs post) and recipient IDs as factor variables. Statistical testing for panel **e** was performed by pathfindR using the hypergeometric test on the results from DESeq2 which were FDR-adjusted for multiple comparisons. $n = 17$ biologically independent transplants for panels (**a**), (**b**) and $n = 12$ biologically independent transplants for panels (**c**) and (**d**). IQR interquartile range (25th–75th percentiles), LOESS locally estimated scatterplot smoothing, MGH Massachusetts General Hospital, PTT post-transplantation time, UCalcium urinary calcium, UCreatinine urinary creatinine, UPhosphorus urinary phosphorus.

data lends some support to this possibility, aldosterone levels were found to be very low suggesting inefficient signal transmission, if any. Despite the very low aldosterone levels post-transplant, measurement of urinary sodium did not demonstrate any persistent concentrating defect. Furthermore, the clinical implications of these findings on human recipients may be minimal. While in preclinical life-sustaining kidney xenotransplantation only the porcine kidney is responsible for initiating and regulating the RAAS pathway, in human xenotransplantation, the recipients' native kidneys would be left in place, as in allotransplantation. Patients with CKD and more importantly, ESKD, often have hyperactive RAAS and take ACE inhibitors or ARBs[1]. Dialysis-dependent ESKD patients have been shown to have no change or only mildly decreased RAAS activity in vivo (compared to CKD patients or healthy controls, respectively)[23,24]. Given the high likelihood for significant residual RAAS function in ESKD patients who retain their native kidneys in a xeno-clinical trial, lack of porcine kidney contribution to the recipient RAAS pathway may be less of a safety concern. However, measurement of blood pressure, RAAS activity, and urinary concentrating ability (urinary sodium and osmolarity) after porcine to human xenotransplantation will be critical to support this hypothesis.

Abnormal calcium and phosphorus levels occurred in every pig to NHP transplant performed. In some cases, these chemistries were severely altered and might be associated with adverse effects in humans, therefore, several differential diagnoses were investigated. Measurement of PTH pre- and post-transplant demonstrated appropriate suppression in the setting of hypercalcemia and no PTHrP was detected. Calcifediol and calcitriol levels were similar to pre-transplant ranges, which ruled out hypervitaminosis-D. In addition, given the relatively short half-life of calcitriol, this suggests adequate hydroxylation by porcine *CYP27B1*. To investigate the possibility of a PTH-like protein being secreted from the porcine kidney and leading to increased bone resorption, CTx was assayed and demonstrated bone resorption was decreased in the presence of hypercalcemia, consistent with nearly absent PTH, and suggesting that bone may not be the source of hypercalcemia.

Given that such prominent hypercalcemia and hypophosphatemia is not a feature of any immunosuppression agent in the regimen and that the recipient signaling axis seems intact, the porcine kidney must be considered the likeliest source. Early after transplantation, urinary calcium decreased and urinary phosphorus increased. However, after 30 days PTT, levels were not significantly different than pre-transplant. In the context of sustained hypercalcemia, this urinary calcium level can be considered inappropriately low, suggesting retention of calcium by the porcine kidney. One possible mechanism involves differential calcium-sensing receptor (CaSR) sensitivity between pigs and primates (amino acid identity ~94% and resting porcine calcium is greater than primates) as in familial hypocalciuric hypercalcemia in humans[25]. The exact mechanism remains unclear but RNA-seq suggested changes consistent with increased cellular machinery to handle chronically elevated calcium (such as *CLDN14* and *CALB1*). While differential CaSR affinity by species is consistent with all

available calcium-PTH-vitamin D axis data, it is not clear how to reconcile the hypophosphatemia observations with this hypothesis. Porcine donors in this study have relatively higher baseline phosphorus levels compared to NHPs (~9 mg/dL; Supplementary Data 1) but xenografted porcine kidneys consistently lead to significant hypophosphotemia in vivo after transplantation. One possibility is that hypophosphatemia stems from proximal tubular dysfunction, as in Fanconi syndrome, perhaps due to a low level of chronic inflammation. Another possibility is that there is differential expression or differential efficiency of a major phosphorus regulating transporter, such as NaPi-IIa (*SLC34A1*) or NaPi-IIc (*SLC34A3*). We can only partially exclude this hypothesis given there is no statistically significant difference in expression of NaPi-IIa in this dataset but NaPi-IIc could not be assessed (*SLC34A3* is located on an unplaced scaffold in Sscrofa11.1 and thus reads are lost during alignment). Finally, it is possible that some unknown compensatory mechanism to hypercalcemia affects phosphorus levels or reabsorption concomitantly. While complete chemistry data after kidney xenotransplantation in NHPs is rarely published, Soin et al. reported borderline hypercalcemia as well as hypophosphatemia in recipients of hDAF porcine kidneys[14]. High normal calcium and hypophosphatemia was also reported in three baboon recipients of more advanced United Therapeutics porcine donors by 3–4 weeks PTT[17]. However, this study is the first to our knowledge to have provided urinalysis, bone investigation, and molecular data to address these phenotypes.

Despite lack of clarity regarding mechanism, these data impact clinical considerations. NHPs used in preclinical studies are healthy while potential human recipients of porcine xenografts will not be. ESKD patients often have hyperparathyroidism from chronic hypocalcemia and hyperphosphatemia. In most cases, renal allotransplantation leads to corrected calcium and phosphorus and normalization of vitamin D and PTH. However, in 15–30% of patients, PTH remains high for >6 months and this can cause persistent hypercalcemia and hypophosphatemia[26–28]. In these patients, a potential interaction exists between a xenospecific PTH-independent hypercalcemia and hypophosphatemia, and a recipient environment with persistently high PTH which has not been modeled in NHPs. Thus, special attention will be required in the postoperative care of recipients with higher pre-transplant PTH. Frequent monitoring and early intervention seems warranted to prevent or reduce the occurrence of severe phenotypes. Calcimimetics such as cinacalcet may be preferable given dual activity on *CaSR* in parathyroid and TAL, though the possibility that some drugs designed for use in humans may not function as expected against non-conserved porcine targets may complicate clinical care (e.g. etelcalcetide (parsabiv) lacks pharmacodynamic effect against porcine *CaSR*)[29].

While this study is based on the largest cohort of long-term surviving NHP recipients after life-sustaining kidney xenotransplantation that we are aware of, some limitations should be noted. All donors were engineered to include 3KO, but different sets of transgenes were expressed in different donors. We do not believe these differences contributed to variation in any of the phenotypes presented, however,

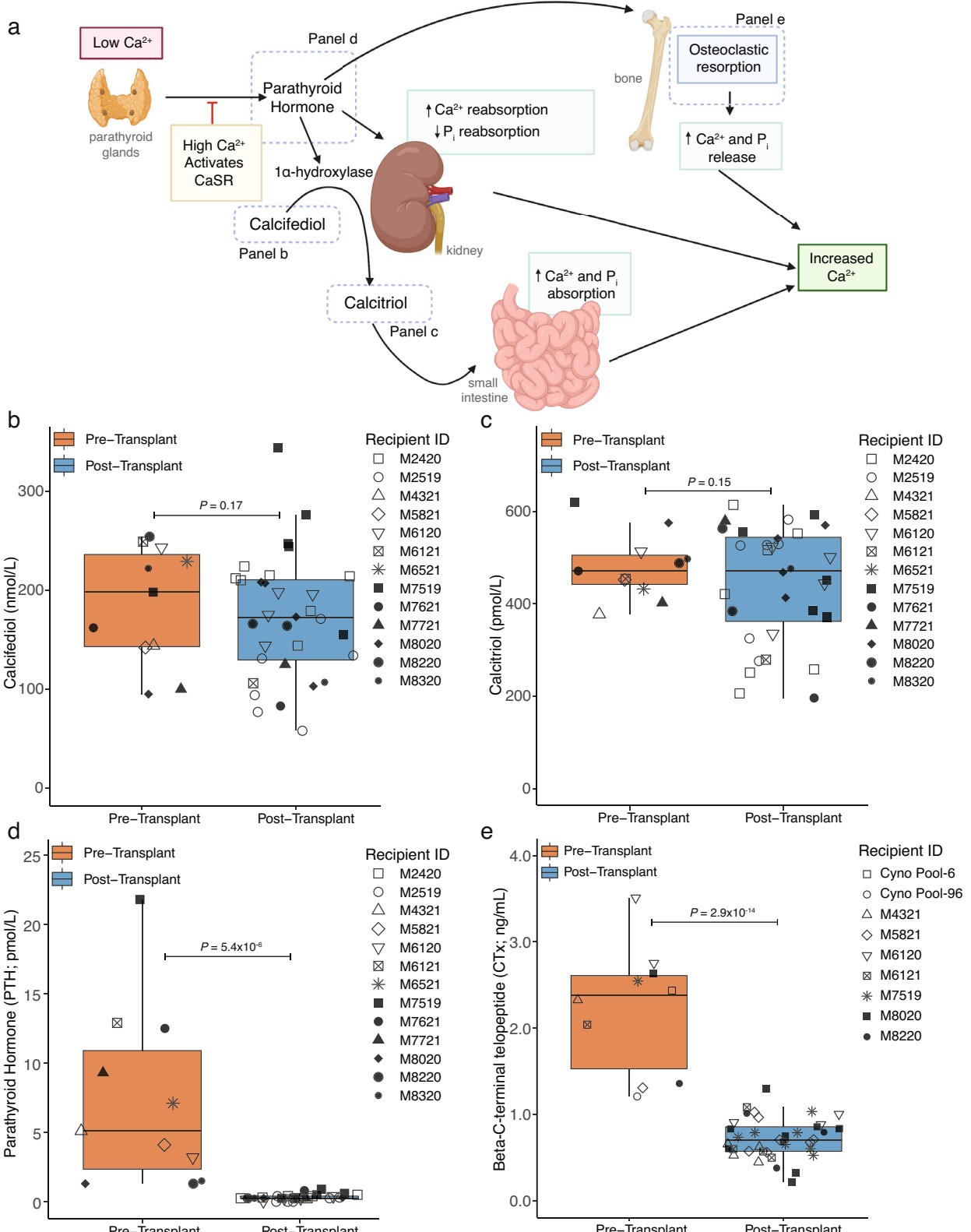

this cannot be entirely excluded. Furthermore, given our interest was in long-term function, we selected recipients with survival >60 days. These animals are part of a larger group of 25 total transplants, 8 of which were euthanized before 60 days. These transplants were excluded due to concerns that processes associated with graft loss may mask the features and physiology associated with long-term function. More importantly, this survival rate may not be representative of human xenotransplantation given certain incompatibilities of transplanting an organ designed for humans into NHPs and challenges in NHP post-transplant care[5]. An additional important role of the kidney, production of erythropoietin, is not addressed in this work due to clinical comfort and readily available pharmacological replacement with agents such as Epoetin alfa and hypoxia-inducible factor prolyl-hydroxylase inhibitor. Another limitation is that there is

**Fig. 5 | Kidney xenotransplantation does not impact vitamin D levels, but alters parathyroid hormone and bone turnover levels. a** Schematic of calcium-parathyroid hormone-vitamin D pathway (created with Biorender.com). **b** Calcifediol was measured in pre- and post-transplant samples from NHP recipients without clear trend observed. **c** As in (**b**) but for calcitriol. **d** As in (**a**) but for PTH which was significantly and rapidly decreased in NHP samples following transplantation. **e** CTx was measured to evaluate the levels of bone resorption before and after transplantation. For box and whisker plots, the central line in each box represents the median of the distribution, i.e., the value that separates the lower 50% of observations from the upper 50%. The box itself represents the

interquartile range (IQR), which spans from the 25th to the 75th percentile of the distribution. The lower bound of the box is the first quartile (Q1), while the upper bound is the third quartile (Q3). The whiskers of the box and whisker plot extend from the box to the minimum and maximum observations within 1.5 times the IQR of the lower and upper quartile, respectively (panels **b–e**). Statistical testing was performed using generalized linear models with xenotransplantation bin (pre vs post) and recipient IDs as factor variables. $n = 13$ biologically independent transplants for panels (**b–d**), $n = 7$ biologically independent transplants for panel (**e**). CaSR calcium sensing receptor, CTx beta-C-terminal telopeptide, IQR inter-quartile range (25th–75th percentiles), PTH parathyroid hormone.

less urine data compared to blood. Collecting NHP urine in a longitudinal fashion is challenging. While sedation and foley catheterization or cystocentesis are possible, foley catheterization introduced unacceptably high rates of UTI[30]. We chose to collect relatively fewer urine samples where the sample was observed and collected immediately, in a nonsterile fashion. While non-sterile collection precludes the use of some types of assays (i.e., sensitive biomarkers or genomics platforms), the urinalysis chemistry datasets can be compared between animals. Finally, while application of modern genomics to xenotransplantation granted insight into the phenotypes observed herein, the tissue collection protocol was designed to observe gross changes in the graft over time and was not a priori designed for study of hypercalcemia/hypophosphatemia. Since the earliest biopsies were around 40–50 days PTT, after establishment of these phenotypes, the transcriptomic differences represent a mix of causative and compensatory changes, which cannot be untangled ad hoc.

In conclusion, this study provides in-depth examination of some of the most important remaining questions prior to FIH clinical trials. In this dataset, we did not observe clinically concerning growth of the xenograft. We have provided data suggesting that the porcine xenograft does not participate efficiently in the recipient RAAS pathway. Finally, in every transplant in this study, hypercalcemia and hypophosphatemia were observed. While several xenograft-extrinsic mechanisms were ruled out, the precise mechanism remains elusive. This represents one of the main safety concerns that will need to be studied in further preclinical work and ultimately monitored and treated in the FIH clinical study.

## Methods

### Animal research study ethical approval and care
The care and treatment of all animals used in these studies were conducted in accordance with the Guide for the Care and Use of Laboratory Animals (Institute of Laboratory Animal Research, National Research Council, U.S. Department of Health and Human Services) and with the approval of the Massachusetts General Hospital Institutional Animal Care and Use Committee (Protocols 2017N000216 and 2017N000214). Adult cynomolgus monkeys (*Macaca fascicularis*) were obtained from Charles River Laboratories, Alpha Genesis, Inc., and BC US LLC. Monkeys were typically socially housed except as necessary for study activities and received High Protein Monkey Diet (LabDiet, 5045) as well as produce and other dietary enrichment. Porcine donors were typically individually housed and received Minipig Grower Diet (LabDiet, 5081). Euthanasia criteria for recipients included serum creatinine of greater than or equal to 6 mg/dL or when a humane endpoint was met as assessed by veterinary staff. Recipients were euthanized with a mixture of pentobarbital sodium and phenytoin sodium.

### Genotype and production of donor animals
Donors were produced and provided by eGenesis. All donors were engineered to eliminate three known glycan xenoantigens using CRISPR-mediated non-homologous end joining to deactivate the enzymes responsible for their generation (3KO: alpha-1,3-galactosyltransferase [*GGTA1*], cytidine monophospho-N-acetylneuraminic acid hydroxylase [*CMAH*], beta-1,4-galactosyltransferase 2 [*B4GALNT2*]). All

kidney xenografts expressed human proteins that regulate important pathways related to xenotransplantation success, including complement, coagulation, and innate immunity. To increase the cohort size of this study, and because a plausible involvement of the different transgenes in the pathways under investigation seemed unlikely, multiple different genotype donors were included in this study (Supplementary Tables 1, 2). Retroviral inactivation of porcine endogenous retroviral elements using CRISPR/Cas9 mediated non-homologous end joining was performed in the case of the clinical candidate genetics, EGEN-2784. Production of gene-edited porcine donors generally involved: gene editing of primary cells from wildtype Yucatan minipigs in vitro, confirming desired genetic alterations, and somatic cell nuclear transfer to generate cloned pigs expressing genes of interest[30,31]. In some cases, serial cloning and editing were conducted to layer edits (e.g., retroviral inactivation after insertion of human transgenes). For this study, all donors were female and were produced by cloning.

### Life sustaining kidney xenotransplantation
On day 0, kidney procurement was performed under general anesthesia using a deceased donor multi-organ protocol as in clinical transplantation including heparin administration and in situ flush with University of Wisconsin preservation solution. For the recipient operation, after establishing appropriate general anesthesia, a midline incision was made and the porcine kidney was transplanted intraperitoneally by anastomosing renal artery and renal vein to the abdominal aorta and vena cava, respectively. Typically, donor warm ischemia time was <1 min, cold ischemia time ranged from 30–60 min and anastomosis time was <30 min. Ureterovesical anastomosis was performed by Lich-Gregoir method without stenting. Bilateral native nephrectomy was performed simultaneously in 13/17 cases. In four cases, due to inadequate urine output or marginal recipient status during surgery, a single native kidney was left in place and later removed around day 20. In 12/17 xenotransplantation cases, central venous access was established by internal jugular vein catheterization on day −5 and removed after the first post-operative week. For the first few weeks postoperatively, the transplanted kidney was monitored by urine output, ultrasound, and serum creatinine measurement twice a week then weekly thereafter.

### Immunosuppression and post-operative management
Induction therapy included anti-CD20 [2B8R1F8]-Afucosylated antibody (NIH Nonhuman Primate Reagent Resource [NIH-NPRR], Cat# PR-8288, RRID:AB_2819341, 20 mg/kg) on day −5, Anti-rhesus thymocyte (rhATG, NIH-NPRR, Cat# PR-0000e, RRID:AB_2716327, 5 mg/kg) on days −1 and 0, anti-CD154 mAb (5C8H1 or TNX-1500, NIH-NPRR, Cat# PR-1547, RRID:AB_2716324 or Tonix Pharmaceuticals, respectively, 25 mg/kg) two doses on day 0, then a single dose of 20 mg/kg on days 2, 5, 7, and 12. Daily solumedrol (Pfizer) was tapered to off during the first 30 days post-transplant and tacrolimus (Astellas) was administered intramuscularly daily for 60 days targeting trough levels of 8–13 ng/mL. Weekly anti-CD154, and daily mycophenolate-mofetil were continued as maintenance therapy (MMF, Genentech; 200 mg/day) [Fig. 1a]. Recipient erythropoiesis was supported by daily, three-times weekly, or weekly exogenous epoetin alfa (Epogen, Amgen) to maintain

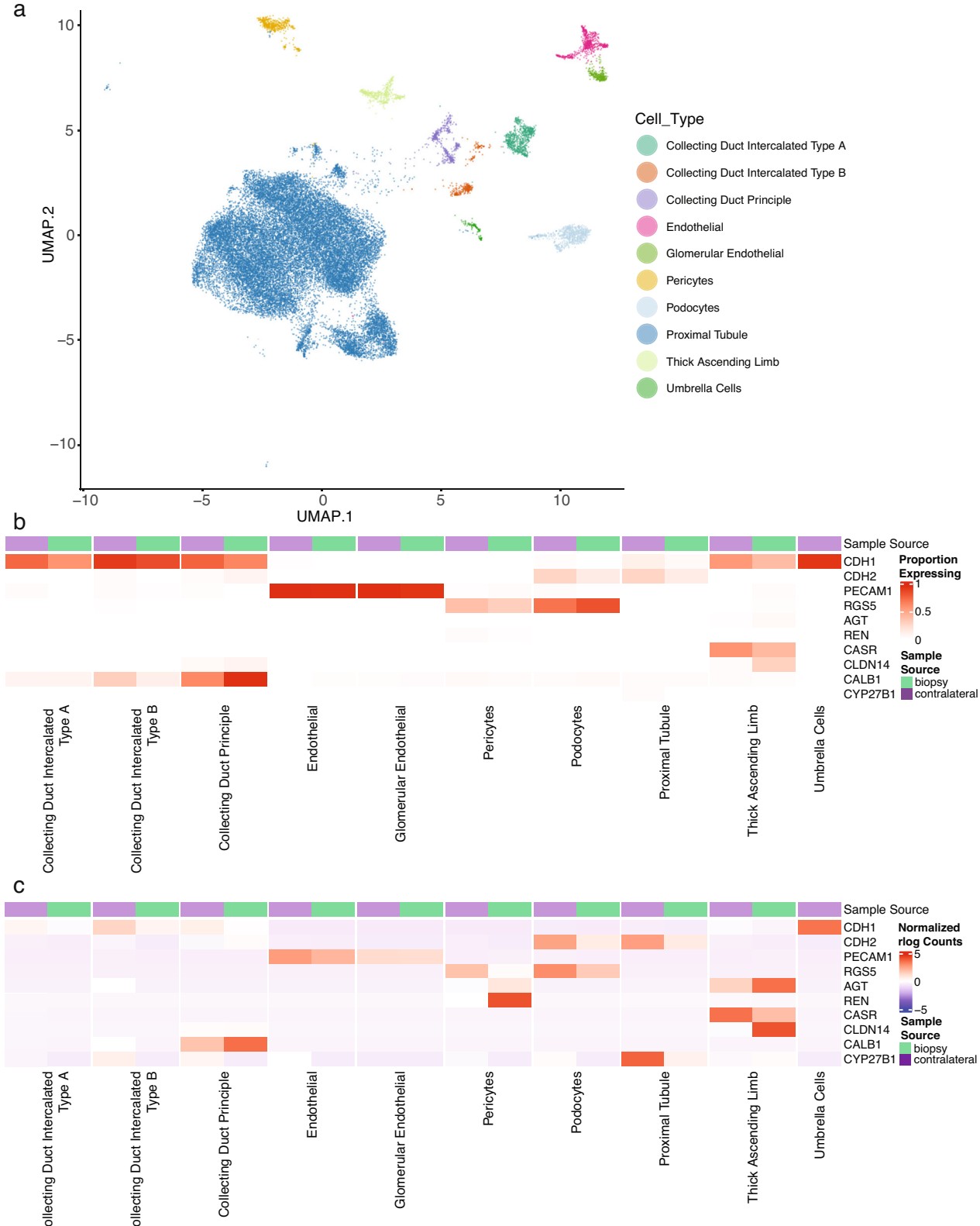

**Fig. 6 | scRNA-seq profiles demonstrate transcriptional changes in xenograft cell compartments. a** UMAP demonstrates 10 distinct cell types after filtering the contralateral and biopsy datasets for porcine parenchymal cells which passed quality control filters and cell identity analysis using prototypical transcripts. **b** Transcriptional heatmap demonstrating the proportion of cell types expressing at least one transcript of interest. **c** As in (**b**) except quantifying the transcript abundance as normalized rlog counts. *n* = 3 biologically independent contralateral untransplantated kidney samples and *n* = 2 biologically independent biopsy tissue samples. scRNA-seq single-cell RNA sequencing, UMAP uniform manifold approximation and projection.

hemoglobin >9.5 g/dL. To minimize the chance of contamination of anti-pig natural antibodies, transfusion was performed only for hematocrit lower than 20%. Prophylaxis included ganciclovir (5 mg/kg) on days 0-90 and enrofloxacin until day 14 or trimethoprim/sulfamethoxazole continued throughout.

Animal weights were obtained weekly and a loss greater than 25% from pre-transplant was an end of study criteria. Clinical pathology (CBC, chemistries) was conducted using Catalyst Dx Chemistry Analyzer (Idexx) and HemaTrue Veterinary hematology analyzer (Heska) with blood collected on days 2, 5, 8, 12, and then every 1–2 weeks or if clinically indicated. Endocrine testing was performed by Michigan State University Veterinary Diagnostic Laboratory and consisted of clinical measurement of aldosterone, calcifediol, calcitriol, ionized calcium, parathyroid hormone (PTH), and parathyroid hormone-related peptide (PTHrP). For urine collection, a clean cotton/polyester mesh was placed over the pan at the bottom of the housing location of each recipient, urine was able to pass through without contamination of feces or food products. The animals were observed until urination, after which the liquid was collected and sent for analysis by Idexx BioAnalytics. In the perioperative period, xenograft ultrasonography was performed using a LogiQe Ultrasound System (GE Healthcare) on days 2, 5, 8, 12 and then every 1–2 weeks to monitor for vascular or ureteral complications and to monitor xenograft size. Xenograft length was measured in the long-axis on cross section and hydronephrosis described by the Society for Fetal Urology (SFU) grading schema[32]. Protocol xenograft biopsies were obtained every 2–4 months in recipients with stable function or for cause if a rise in serum creatinine occurred. Tissue was processed for light microscopy and another portion processed for RNA-seq as described below. At end of study, a complete necropsy was performed for histopathologic examination of the renal xenograft, lymph nodes, heart, lung, liver, pancreas, thymus, and skin. Parathyroid and thyroid tissue was obtained in the case of one long-term survivor (M2519). Additional tissue from the xenograft was processed for RNA-seq at necropsy.

### Renin activity (plasma and serum) assays
The plasma renin activity (PRA) and serum renin activity (SRA) assays were based on previously reported methods[33]. Briefly, citrated plasma or serum was mixed with 8-hydroxquinolone hemi-sulfate and EDTA and incubated for 0 or 6 h at 37 °C before the reaction was halted with BSA buffer resulting in a final dilution of 1:20. Production of Angiotensin-I (AngI) was measured using an Angiotensin-I ELISA kit (Novus Biological, NBP 262134) according to the manufacturer's instructions with absorbance read on a FilterMax F5 microplate reader (Molecular Devices). Values below the level of detection were considered as zero pg/mL. PRA or SRA was calculated as AngI at 6 h – AngI at 0 h divided by length of incubation, e.g., pg/mL/6 h. Activity was not adjusted for dilution associated with the addition of reaction buffers.

### Beta-C-terminal telopeptide (CTx) assay
NHP recipient serum pre- and post-transplant and pooled cynomolgus serum were diluted 1:5 and run in the Serum CrossLaps® beta-C-terminal telopeptide (CTx) ELISA (Immunodiagnosticsystems, AC-02F1) according to the manufacturer's instructions with absorbance read on a FilterMax F5 microplate reader (Molecular Devices).

### Isolation of RNA from porcine kidneys and Bulk Illumina RNA-sequencing (RNA-seq)
Samples were collected, snap-frozen, and cryopreserved from contralateral untransplanted kidneys (CUK) at time of organ procurement, biopsies at time of protocol biopsy or kidney at necropsy. On the day of protocol biopsy collection, the mean serum creatinine value was 0.7 mg/dL and mean blood urea nitrogen was 22.9 mg/dL indicating normal renal function. Sequencing runs were minimized to reduce batch effect. For a given batch, snap-frozen tissue was pulverized using

a pre-chilled BioPulverizer (BioSpec, 59012MS). Total RNA was extracted using the Qiagen RNeasy kit (Qiagen, 74106) after passing the sample through homogenizing columns (Qiagen, 79656) and performing a DNAase digestion (Qiagen, 79256) according to manufacturer's instructions. RNA integrity number and concentration were measured using an automated electronic electrophoresis system (Agilent, 4200 Tapestation) and reagents (Agilent, 5067-5578). For those samples passing quality checks, 500 ng of total RNA was carried into the Illumina Stranded mRNA kit library preparation (Illumina, 20040534). Libraries were characterized using the high-sensitivity dsDNA assay (Invitrogen, Q33231) for Qubit 3 (Invitrogen, Q33216). Libraries of appropriate concentration and insert length were diluted to loading concentration of 650 pM and loaded and sequenced using a P3 100 cycle kit (Illumina, 20040559) on Illumina NextSeq 2000.

### Bioinformatic processing and computational analysis of Bulk Illumina RNA-sequencing
ENSEMBL Sscrofa11.1 FASTA assembly and v105 GTF annotation were prepared as a reference by removing unplaced contigs/scaffolds, and then adding intended genetic alterations/payload sequences as a separate contig. ENSEMBL Macaca fascicularis 6.0 FASTA assembly and v105 GTF annotations with unplaced contigs/scaffolds removed were appended to the pig reference for aligning reads from samples with NHP transcripts. Transcript sequences were generated with *gffread* (0.12.7)[34] on these synthetic genome assemblies. Indexing was performed on transcripts with *salmon index* (1.6.0)[35] applying standard settings and the whole genome as a decoy. *Salmon quant* in mapping-based mode was used to quantify RNA-seq read counts per transcript using custom settings: 100 bootstrap replicates, GC content and sequence bias correction applied. For this study, only transcripts aligning to Sscrofa11.1 were retained and analyzed. Gene level aggregation on transcript abundance was performed with *tximport* (1.22.0)[36]. Protein coding and lncRNA genes were retained and normalized rlog counts calculated using *DESeq2* (1.36.0)[37] with design including sample source (CUK, biopsy, necropsy) and pig identification number used as covariates. Heatmaps were created using *ComplexHeatmap* (2.12.1)[38]. Pathway analysis was performed using *pathfindR* (1.6.4) on KEGG and GO pathways[39].

### Single-cell isolation for single-cell RNA-sequencing (scRNA-seq)
Dissociation of kidney samples into single cell suspensions was performed using dissected cortex from necropsy and CUK samples or from approximately half of a biopsy sample. Tissue was minced into 1–2 mm³ pieces and digested with type IV collagenase (Sigma, C5138-500 mg; 1 mg/mL in 10 mL of Hank's Buffered Salt Solution (HBSS) [ThermoFisher, MT-20023CV]). The resulting single-cell suspensions were passed through a 100 μm cell strainer (Miltenyi Biotec, 130-098-463), centrifuged and then passed through a 70 μm cell strainer (Miltenyi Biotec, 130-098-462) and centrifuged again. Red blood cell (RBC) lysis was performed by resuspending these pellets in 4 mL RBC lysis buffer (Roche, 11814389001) and placing on ice for 2 mins RBC lysis was quenched by adding 25 mL 20% fetal bovine serum in HBSS followed by centrifugation and the pellets resuspended in HBSS and filtered through 70 μm filters. A dead cell removal step was performed using a Dead Cell Removal Kit following the manufacturer's instructions (Miltenyi Biotec, 130-090-101). After dead cell removal, cells were counted and resuspended at ~600–1200 cells/μL. These suspensions were applied to the 10X Genomics Chromium Next GEM Single Cell 3' v3.1 kit to prepare scRNA-seq libraries, per manufacturer's instructions (10X Genomics, 1000121).

### Bioinformatic processing and computational analysis of 3' scRNA-seq
Assembly construction and annotation were as described for bulk RNA-seq and was indexed here using *STARsolo* (2.7.9a) *genomeGenerate* on

standard settings[40]. 10 × 3′ scRNA-seq was quantified using custom settings for STARsolo –soloCellFilter EmptyDrops_CR, --soloMulti-Mappers EM, --clipAdapterType CellRanger4, --outFilterScoreMin 30, --soloCBmatchWLtype 1MM_multi_Nbase_pseudocounts, --soloType Gene, --soloUMIdedup 1MM_CR, --soloUMIfiltering MultiGeneUMI_CR, and –soloUMIlen 12. Counts were read into R (4.2.0) with *DropletUtils* (1.14.2)[41] and cell quality checking and control performed using a set of filtering criteria: number of unique molecular identifiers (UMI), detected features, %mitochondrial UMI, %protein-coding and lncRNA UMIs, complexity, and doublet removal using *scDblFinder* (1.8.0)[42]. Count normalization was performed using *scran* (1.22.1) per recommendations for quickCluster, computerSumFactors, and logNormCounts[43]. Cell-cycle analysis and annotation was performed with *Seurat's* (4.1.0) *CellCycleScoring*[44]. *scVI* was used to model and remove batch effect using covariates: %mitochondrial UMI, Seurat Cell Cycle S and G2M phase scores, pig donor identification, and kidney region (i.e., cortex vs medulla)[45]. The 20 latent dimensions from *scVI* were passed to scran's *clusterCells* and clustering performed using the *bluster* (1.4.0) Louvain algorithm on shared nearest neighbors (k = 3; https://www.bioconductor.org/packages/release/bioc/html/bluster.html). Uniform manifold approximation and projection (UMAP) was created via *scater's* (1.22.0) *runUMAP* with 20 nearest neighbors and minimum distance 0.2[46]. Cell types were assigned comparing known markers[47] and the molecular signature database[48] cell type signature enrichment from *fgsea* (1.20.0; for a function on top 50 features per cell cluster)[49] against cluster marker genes from *scran's scoreMarkers*. UMAP plots were generated using *scater's ggcells* and *ggplot2* (3.2.0).

## Statistics

For clinical data, generalized linear models were constructed with appropriate distribution (gaussian, negative binomial, or zero adjusted gamma) in *gamlss* (5.4.12) using *glm*. Fixed effects models were constructed by adding dummy variables for recipient-ID to models of xenograft length over time using *glm*. For RNA-seq, *P*-value adjustment for multiple comparisons was performed using the FDR method described by Benjamini-Hochberg and implemented in *DESeq2*. All analyses were performed using RStudio 2022.02.2 Build 485, which interfaces with R (4.2.0) and tests were two-tailed using an alpha of 0.05.

## Reporting summary

Further information on research design is available in the Nature Portfolio Reporting Summary linked to this article.

## Data availability

Bulk and single-cell RNA sequencing data that support the findings of this study have been deposited in NCBI Sequence Read Archive (SRA) and the Gene Expression Omnibus (GEO) and can be found under GSE 216034, GSE210556, and GSE210557. For a subset of transplants, both bulk and scRNAseq are contained in GSE216034. For the remainder of the transplants, bulk RNAseq is contained in GSE210556 and scRNAseq in GSE210557. Molecular Signatures Database (MsigDB) is available at https://www.gsea-msigdb.org/gsea/msigdb/. The KEGG database is available at https://www.genome.jp/kegg/pathway.html. The GO database is available at http://geneontology.org/. ENSEMBL Sscrofa 11.1 and ENSEMBL Macaca fascicularis 6.0 FASTA assemblies are available at https://www.ensembl.org/index.html. Additional data underlying the figures is available in the source data file. Source data are provided with this paper.

## Code availability

Unique code used in the generation of this study has been deposited in github and can be accessed at https://github.com/egenesis/Xenokidney-Physiology-Nature-Communications.

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

## Acknowledgements

We are grateful for the efforts of our colleagues at eGenesis whose hard work led to the generation of gene-edited porcine donors and creation of the bulk RNAseq and single-cell RNAseq dataset used in this study. The anti-CD20, rhATG, and anti-CD154[5C8H1] were provided by the NIH Nonhuman Primate Reagent Resource (ORIP P40 OD028116, NIAID U24 AI126683). We thank Tonix Pharmaceuticals for providing TNX-1500. We appreciate Rajesh Thakker's (University of Oxford) and Michael Mannstadt's (Harvard Medical School) feedback on the work contributing to this manuscript. Figures 1A, 3A, and 5A were created with Bior-ender.com. D.J.F. received an AST Translational Research Fellowship which partially supported this work. D.J.F. and G.L. were supported by 5T32AI007529 from NIH. eGenesis provided funding for xenotransplant experiments.

## Author contributions

G.L., T.H., A.D., and T.K. performed transplants and provided post-transplant care. D.J.F., R.P., and K.C.H. performed RNA-seq, scRNA-seq, in vitro assays, and computational analysis. D.J.F., K.C.H., and G.L. conceived the study. D.J.F., G.L., T.H., R.P., J.F.M., T.K., and K.C.H. critically evaluated the analyses. D.J.F. and K.C.H. drafted the manuscript. All authors critically reviewed and edited the manuscript and approved its final form.

## Competing interests

D.J.F. received fellowship salary support from eGenesis. K.C.H. and R.P. contributed to this work as employees of eGenesis, Inc. and have equity interest, in the form of stock options, in eGenesis, Inc. eGenesis has filed multiple patent applications covering the subject matter of this manuscript. G.L., J.F.M., and T.K. have served as consultants to eGenesis. The remaining authors declare no competing interests.
