## [Peer Review File · Nature Communications]

Clinical and molecular correlation defines activity of physiological pathways in life-sustaining kidney xenotransplantationEditorial Note: This manuscript has been previously reviewed at another journal that is not operating a transparent peer review scheme. This document only contains reviewer comments and rebuttal letters for versions considered at *Nature Communications*.

REVIEWERS' COMMENTS

Reviewer #1 (Remarks to the Author):

The reviewers addressed my comments appropriately. This represents a tremendous amount of work and expertise with a relatively raw presentation of the various physiologic findings, which will undoubtedly be helpful for the field of transplantation and beyond.

The concern of the FDA about kidney size is understandable. However, it seems that the longitudinal analysis of xenograft length remains superficial and not at the same depth level as other components presented here. Maybe the authors want to consider multiple imputations for the missing data or management of outliers (or present the data points with mean rather than lines). The lines presented as it presently is in Graph B of Figure 2 are difficult to follow (i.e., the actual kidney length variation is most likely continuous and relatively smooth. The ups and downs shown here most likely represent assessment method variability.).

Also, it might be advisable to clearly state that all physiologic variations observed here (RAA, Calcium, Pi, etc.) might have been due to some over-compensation of other mechanism activation/deactivation specific to the xenotransplantation context and not identified/observed/studied here.

Reviewer #2 (Remarks to the Author):

I have closely read the detailed responses to my questions that I find have been addressed appropriately. Authors have also expanded their work within a feasibility scope and time limit, underpinning most of the relevant issues brought up by the reviewers.

Reviewer #3 (Remarks to the Author):

The authors have adequately addressed the concerns of this reviewer.

REVIEWERS' COMMENTS

Reviewer #1 (Remarks to the Author):

The reviewers addressed my comments appropriately. This represents a tremendous amount of work and expertise with a relatively raw presentation of the various physiologic findings, which will undoubtedly be helpful for the field of transplantation and beyond.

The concern of the FDA about kidney size is understandable. However, it seems that the longitudinal analysis of xenograft length remains superficial and not at the same depth level as other components presented here. Maybe the authors want to consider multiple imputations for the missing data or management of outliers (or present the data points with mean rather than lines). The lines presented as it presently is in Graph B of Figure 2 are difficult to follow (i.e., the actual kidney length variation is most likely continuous and relatively smooth. The ups and downs shown here most likely represent assessment method variability.).

Also, it might be advisable to clearly state that all physiologic variations observed here (RAA, Calcium, Pi, etc.) might have been due to some over-compensation of other mechanism activation/deactivation specific to the xenotransplantation context and not identified/observed/studied here.

Author Response:

We thank the reviewer for their time and effort. We appreciate their input and believe that it has resulted in a stronger manuscript.

The analysis of the kidney xenograft included only one type data (measurement by US); whereas for endocrine pathways, we were able to measure multiple analytes and use orthogonal modes of analysis. This may have led to the reviewer's observation about relative depth of analysis on that topic. However, we did not believe that additional available data (RNA sequencing analysis) or other possible modes of measurement (CT scan) would fundamentally change or enrich our analysis. Therefore, we have limited our data presentation to the current version. In terms of data presentation, we have included the LOESS line in this graph and highlighted its use in the figure legend. The LOESS line incorporates all the values from each of the transplant resulting in clearer communication of the finding of minimal growth overtime.

Regarding the reviewer's last point, we agree that the observed phenotypes could result from unconsidered mechanisms. We believe that our current discussion is nuanced enough to encompass this possibility.

Reviewer #2 (Remarks to the Author):

I have closely read the detailed responses to my questions that I find have been addressed appropriately. Authors have also expanded their work within a feasibility scope and time limit, underpinning most of the relevant issues brought up by the reviewers.

Author Response:

We thank the reviewer for their time and effort. We appreciate their input and believe that it has resulted in a stronger manuscript.

Reviewer #3 (Remarks to the Author):

The authors have adequately addressed the concerns of this reviewer.

Author Response:

We thank the reviewer for their time and effort. We appreciate their input and believe that it has resulted in a stronger manuscript.